# Testing Leash Walking Training as a Physical Activity Intervention for Older Adult Dog Owners: A Feasibility Study

**DOI:** 10.3390/geriatrics7060120

**Published:** 2022-10-24

**Authors:** Katie Potter, Caitlin Rajala, Colleen J. Chase, Raeann LeBlanc

**Affiliations:** 1Department of Kinesiology, University of Massachusetts Amherst, Amherst, MA 01003, USA; 2Elaine Marieb College of Nursing, University of Massachusetts Amherst, Amherst, MA 01003, USA

**Keywords:** aging, dog walking, human–animal bond, animal-assisted intervention, sedentary behavior

## Abstract

Dog walking is a physical activity (PA) with many health benefits for older adults. Dog behavior issues can be a barrier to dog walking. This study piloted leash manners training as a PA intervention for dog owners ages 60+ years. Fourteen dog owners (mean age = 65 years, female = 71%) enrolled in a leash manners training course. Process evaluation data were collected to determine feasibility and acceptability of the intervention and study procedures. Twelve of fourteen enrolled participants (86%) completed the course, and most were highly engaged with the program, as evidenced by high class attendance (92% of participants who completed the course attended ≥5 of 6 classes) and a majority (64%) reporting frequent skills practice at home. Further, most participants (73%) reported their leash walking skills improved. The PA assessment protocol (7 days of 24 h data collection using activPAL monitors) was well tolerated, with all participants who completed assessments at each time point (pre-program, post-program, 4-week follow-up) providing ≥6 valid days of data. In sum, the intervention approach and study procedures were feasible and acceptable in this sample of older adults. More research is needed to determine the effectiveness of leash manners training for increasing PA in this population.

## 1. Introduction

Regular physical activity (PA) facilitates healthy aging. Leading a physically active lifestyle reduces the risk of cardiovascular disease, type II diabetes, depression, and various cancers [1]. It also helps with the management of these conditions, supports maintenance of physical and cognitive function, and reduces the risk of falls and fall-related injuries [1]. Maintaining health and functional ability are key predictors of life satisfaction in older adulthood [2]. An estimated 38% of older Americans are physically inactive, defined as reporting no aerobic PA of at least 10 continuous minutes in a typical week [3]. An additional 22% are ‘insufficiently’ active, defined as doing some aerobic PA, but not enough to meet the minimal aerobic PA guideline of 150 min per week [3]. Although a number of factors beyond the reach of health behavior scientists affect PA levels in older adults (e.g., level of education, income level [4]), there is still a need for behavioral interventions with high-level reach that motivate and support PA among older adults. 

One novel approach for combating low PA levels in this population may be to support dog walking. Nearly one in two Americans in their 60s, and more than one in three Americans ages 70+, own a dog [5]. Older adult dog owners who walk their dogs are more physically active and have better physical health and function than those who do not walk their dogs [6,7,8,9]. They may also spend more time outdoors [10] and interact more with neighbors [11]. Yet, studies suggest that 52–64% of older adult dog owners do not engage in dog walking regularly [7,12]. Notably, no dog walking interventions to date have targeted community-dwelling older adults [13].

Curl et al. found that 40% of older adult dog owners who reported not walking their dog cited dog behavior as the reason they did not walk, while only 6% cited lack of time or interest [6]. Thus, the purpose of this study was to pilot a leash walking training course enhanced with behavior change strategies as a PA intervention for older adult dog owners. Improving a dog’s behavior on leash may make dog walking more enjoyable, and enjoyment is key to PA adherence [14,15]. Obedience training may also strengthen the bond between dog and owner [16,17], a correlate of dog walking behavior [6]. Finally, and especially important for this population, improving a dog’s behavior on leash may reduce the risk of falls and fall-related injuries while dog walking [18].

In this study, we piloted a 6-session leash manners training course enhanced with behavior change strategies as a PA intervention for dog owners ages 60+ years. We collected process evaluation data to determine the feasibility and acceptability of the intervention approach and study procedures, including all data collection procedures.

## 2. Materials and Methods

### 2.1. Study Design and Population

This study was a single-arm, proof-of-concept trial with measures at baseline, post-program (four weeks), and follow-up (eight weeks). Community-dwelling older adult dog owners (ages 60+ years) who reported dog walking ≤75 min per week, other aerobic activity ≤3 days per week, and who could walk unaided for ≥10 min continuously were eligible to participate. To minimize fall risk in this initial study, individuals who reported ≥2 falls in the last year, sought medical attention for a fall, or who reported feeling unsteady while walking were excluded [19]. Additional fall risk screening included the Mini-Cog screener [20] and Timed Up-and-Go (TUG); individuals were excluded if they scored <4 on the Mini-Cog or took ≥12 s to complete the TUG [21]. Finally, individuals were excluded if they owned a dog who could not walk for at least 10 min at a time or who exhibited past aggressive behavior. All participants provided informed consent. This study was approved by the University’s Institutional Review Board and the Institutional Animal Care and Use Committee.

### 2.2. Procedures

Participants were recruited through social media and a University press release that led to local media coverage, as well as through community outreach (e.g., YMCA, senior centers, farmer’s markets). Eligible individuals enrolled in a leash manners training course enhanced with health behavior change strategies. Classes met twice per week for three weeks for 45 min per class. Classes were held on Wednesday evenings and Sunday mornings, with seven students per class. Classes were led by a certified behavior adjustment trainer who uses only positive reinforcement training techniques. Classes were held in a large, all-purpose room and outdoors in a grassy area on the University campus from 4–22 September 2019.

The initial class session included an introduction to the course by the dog trainer as well as a short presentation by the principal investigator that discussed the health benefits of PA, the aerobic PA guidelines, the goal of the program (‘to help make dog walking a safe, enjoyable part of your daily routine’), and strategies for creating a dog walking habit (event-based cue selection, consistent performance, tracking). The habit formation instruction was guided by the Multi-Process Action Control (M-PAC) Framework [22,23,24,25]. Classes 2–6 involved dog training only (i.e., no additional behavior change information was delivered). Skills covered included how, when, and why to positively reinforce the dog for specific behaviors (e.g., checking in with owner, staying close enough to owner to prevent tension on leash, moving with owner and stopping when owner stopped). A brief tutorial on canine body language focusing on stress signals was included, with strategies for moving out of stressful situations explained and demonstrated. Participants received a ‘homework’ handout from the dog trainer after each class to encourage skills practice at home. Participants were also encouraged to track their daily dog walking and training skills practice on a sheet provided by the researchers (see Figure 1).

Device-measured PA and sedentary behavior, dog walking behavior, dog walking enjoyment, and the dog-owner relationship were assessed at baseline, week 4 (post-program) and week 8 (follow-up). Dog walking habit strength (automaticity) was collected at weeks 4 and 8. Participants received a modest financial incentive ($20) at the completion of each assessment.

### 2.3. Measures

#### 2.3.1. Process Evaluation

Feasibility and acceptability of the program were determined by response (number of completed screening questionnaires), recruitment rate (% of completed screening questionnaires enrolled in study), retention rate (% of enrolled who completed week 8 assessments), activPAL protocol acceptability (% providing usable data at each time point, average number of valid wear days at each time point), overall data completeness (% providing valid survey, log booklet, and activity monitor data at all timepoints), participant engagement (class attendance, % reporting skills practice outside of class), and participant feedback (% reporting leash walking skills improved; % who would recommend the class to others).

#### 2.3.2. Physical Activity & Sedentary Behavior

ActivPAL3 micro monitors (PAL Technologies, Glasgow, Scotland) were used to assess PA and sedentary behavior. The activPAL is a small (23.5 × 43 × 5 mm), lightweight (9.5 g), thigh-worn device that uses accelerometer-derived information about thigh position to estimate time spent in different body positions (i.e., sitting/lying, standing, and stepping) and details of PA (e.g., daily steps, cadence) and sedentary behavior (e.g., duration of sedentary bouts) [26,27]. Participants were asked to wear the activPAL 24 h per day for 7 days at each time point. To be included in analysis, participants had to meet valid day criteria (determined by activPAL algorithm) for ≥4 days, including ≥l weekend day.

#### 2.3.3. Dog walking Behavior

Participants logged the start and end time of all dog walks in a paper log booklet on the same days they wore the activPAL monitor. Total walks per week and total days per week with at least one dog walk were calculated.

#### 2.3.4. Dog-Owner Relationship

The Comfort from Companion Animals Scale (CCAS) was used to assess participants’ attachment to their dog in terms of the perceived level of emotional comfort they receive from their dog [28]. The CCAS questionnaire contains 11 items (e.g., “My pet provides me with companionship”, “My pet makes me feel loved”) with response options ranging from strongly disagree to strongly agree on a 4-point Likert scale. Scores range from 0–44, with higher scores indicating greater attachment.

#### 2.3.5. Dog Walking Enjoyment

The Physical Activity Enjoyment Scale (PACES) [29] was used to assess dog walking enjoyment. The 18-item PACES questionnaire asks participants to rate how they feel about the PA they have been doing using a 7-point Likert scale (e.g., “I enjoy it/I hate it”, “I feel bored/I feel interested”). For this study, participants were prompted to focus on dog walking specifically (rather than general PA) when responding to the PACES items. Scores range from 18–126, with higher scores indicating greater enjoyment.

#### 2.3.6. Dog Walking Habit Strength

Four questions from the Self Report Habit Index [30] were used to assess the degree to which participants performed dog walking automatically. Using a 5-point Likert scale ranging from strongly disagree to strongly agree, participants indicated the extent dog walking is something they (1) ‘do automatically’, (2) ‘do without having to consciously remember it’, (3) ‘do without thinking’, and (4) ‘start doing before I realize I’m doing it’. Responses were summed to generate total scores ranging from 4–20, with higher scores indicating stronger dog walking automaticity.

### 2.4. Statistical Analysis

Process evaluation data were summarized using frequencies and percentages. Means and standard deviations were used to summarize PA (steps per day), sedentary behavior (minutes per day spent sitting), dog walking frequency (total walks per week, total dog walking days per week), dog walking enjoyment, dog walking habit strength, and dog-owner bond strength at each timepoint. Means and standard deviations were also used to summarize 4-week and 8-week change scores. Change scores were calculated by subtracting baseline values from week 4 and week 8 values (i.e., week 4 value—baseline value= 4-week change score; week 8 value—baseline value= 8-week change score). Given the preliminary nature of the study, inferential statistics were not performed.

## 3. Results

### 3.1. Participants

See Table 1 for sample characteristics. Of 14 participants enrolled in the study, 86% were in their 60s, 71% were female, and 93% were White/European. The majority had obesity (71%) but reported their general health to be good or very good (93%). Most (86%) lived in rural or suburban communities.

### 3.2. Process Evaluation

A study flow chart is shown in Figure 2. Eighty individuals completed the screening questionnaire and fourteen were enrolled in the study for a recruitment rate of 18%. Eleven participants were retained at final assessments for a retention rate of 79%. The activPAL monitor protocol (7 days of 24 h data collection) was well tolerated, with 100% of participants who completed assessments at each time point (n = 14 pre-program, n = 12 post-program, and n = 11 at 4-week follow-up) providing ≥6 days of valid data. Average number of valid wear days were 6.7 ± 0.5, 7.4 ± 0.8, and 7.3 ± 2.1 at pre-program, post-program, and follow-up, respectively. Ten participants (71%) provided complete data (i.e., activPAL, activity log, and survey data) at all time points. In terms of program engagement, 12 of 14 enrolled participants (86%) completed the 3-week course; 11 of 12 participants who completed the course (92%) attended at least 5 of 6 classes. Of 11 participants who completed the post-program (4-week) survey, 7 of 11 (64%) said they practiced their leash walking skills most or every day of the week during the program. In terms of program satisfaction, 8 of 11 (73%) reported their leash walking skills improved “quite a bit” due to the training, and 8 of 11 (73%) said that they would “definitely” recommend the class to others.

### 3.3. Physical Activity and Sedentary Behavior

Mean steps per day and sitting minutes per day at baseline, post-program (4-weeks), and follow-up (8-weeks), as well as mean change scores at each time point, are reported in Table 2. There was a mean change of 157.8 ± 1125.3 steps per day post-program and 93.0 ± 1594.4 steps per day at follow-up. There was a mean change of −7.0 ± 53.3 sitting minutes per day post-program and −36.1 ± 88.7 sitting minutes per day at follow-up. Figure 3 shows individual changes in mean steps and sitting minutes per day at each time point.

### 3.4. Dog Walking Behavior

Dog walking frequency data from log booklets at baseline, post-program, and follow-up, as well as change scores at each time point, are reported in Table 2. The mean change in days per week with at least one dog walk was 0.8 ± 1.2 post-program and −0.3 ± 1.2 at follow-up. The mean change in total dog walks per week was 0.8 ± 1.7 post-program and −0.9 ± 2.4 at follow-up.

### 3.5. Dog-Owner Relationship, Dog Walking Enjoyment, and Dog Walking Habit Strength

Mean CCAS and PACES scores at baseline, post-program, and follow-up, as well as change scores at each time point, are reported in Table 2. There was virtually no change in the dog-owner relationship as measured by the CCAS from baseline to post-program (mean change 0.0 ± 2.6) or follow-up (mean change −0.2 ± 2.0). The mean change in dog walking enjoyment as measured by the PACES at post-program was 4.2 ± 9.1 and 3.8 ± 11.8 at follow-up. The mean dog walking automaticity (i.e., habit strength) score was 11.6 ± 3.6 at post-program and 11.6 ± 3.7 at follow-up (on a scale of 4–20).

## 4. Discussion

This study piloted a leash manners training course enhanced with evidence-based behavior change content as a novel PA intervention approach for older adult dog owners. Our process evaluation found this approach was feasible and acceptable. Our retention rate was high and most participants were highly engaged with the program, as evidenced by high class attendance and a majority reporting frequent skills practice at home. Further, most participants reported their leash walking skills improved and that they would recommend the course to others. Compliance with data collection procedures was also high, with all participants who completed assessments at each time point providing ≥6 days of valid activity monitor (activPAL) data, and most providing complete data (i.e., activPAL, activity log, and survey data) at all time points. Due to the small sample size and lack of a comparison group, it is not appropriate to draw conclusions about the effects of this intervention based on this study. 

The rationale for this study, from a theoretical perspective, was that improving leash walking skills may make dog walking more enjoyable and strengthen the dog-owner relationship. Notably, we did not observe even modest changes in either proposed mechanism in this study. Participants reported high levels of dog walking enjoyment and attachment to their dog at baseline, and therefore there was little room for improvement. Ceiling effects due to selection bias may be a particular challenge in human–animal bond research studies, as they likely attract participants who are highly attached to their pets and enjoy spending time with them. It is also possible that the duration of the study was too short to observe change in these outcomes, particularly pet attachment. Based on our observations in this pilot study, future studies testing this intervention approach might consider targeting new dog owners who have not yet developed a strong bond with their dog and/or testing a longer program. 

Physical activity is partially regulated by non-conscious processes, including habits [31], and these processes may be especially important for long-term PA maintenance [23]. Further, dogs (like people) flourish with structure and routine [32], which may make dog walking an ideal form of PA for habitual performance. Participants in this study were encouraged to pair dog walking with a consistent, event-based cue (e.g., their morning coffee) and to track their dog walks to facilitate a habitual dog walking routine. The amount of time needed to convert a conscious, effortful choice to a non-conscious, automatic response to a stimulus (i.e., to form a habit) will vary based on the person and the behavior. For example, a recent study found that the successful formation of a nutrition habit ranged from 4 to 335 days, with a median of 59 days [33]. The goal of this program was to set the groundwork for habitual initiation of dog walking even if we were not able to capture it in our assessments. Future dog walking interventions should similarly use an approach based in basic behavior science to encourage habit formation and follow participants over a longer period to determine if and when the initiation of dog walking becomes habitual. 

We chose the activPAL monitor over other PA measurement devices because it provides precise estimates of both PA (steps per day) and daily sedentary time [26,27]. Older adults are the most sedentary age-group [34] and high levels of sedentary behavior negatively impact physical, mental and social well-being and increase mortality risk [35,36,37]. We wanted to collect high-quality sedentary behavior data, as we feel it is plausible that more time spent practicing skills and/or playing with one’s dog after attending training classes could lead to less time sitting, irrespective of any changes in dog walking behavior. Further, modest decreases in device-measured sedentary behavior were observed in another study that tested obedience training as a behavioral health intervention for dog owners [38]. Many participants in this pilot decreased their sedentary behavior post-program and at follow-up. These changes cannot be attributed to the intervention due to the small sample and single-arm, pre-post design (i.e., high likelihood of confounding), but we believe this type of intervention is a plausible pathway for targeting sedentary behavior and that the impact of dog obedience training on this critical health behavior should be explored further.

To our knowledge, no published dog walking interventions have targeted community-dwelling older adults [13]. The only study that has leveraged the human-dog bond to promote PA in older adults was a 2015 pilot study conducted with individuals with cognitive impairment living in assisted-living facilities (Friedmann et al., 2015). Other published dog walking interventions focused on adult populations have taken various approaches, including focusing on the dog’s exercise needs (rather than the owner’s) [39], encouraging dog walking among new dog adopters [40], and encouraging family PA with the dog [41]. A pilot study by Potter et al. that examined obedience training as a PA intervention for sedentary dog owners found that participants randomized to a six-week basic obedience training course slightly increased daily steps post-program compared to a wait-list control group who slightly decreased daily steps, for an average group difference of 780 steps per day [38]. Group differences persisted six weeks later (average difference 1084 steps per day) [38].

This preliminary investigation was a single-arm study with a small sample size and therefore inferential statistics were not performed due to the high risk of uncontrolled confounding. We reported individual participant changes in PA and sedentary behavior in addition to group averages to demonstrate the high variability in change in these outcomes. While some participants had clinically meaningful improvements in both outcomes, others took fewer steps and sat more minutes per day after participating in the intervention. There are many factors that can affect PA and sedentary behavior week to week (e.g., weather, acute health status, variations in work and family responsibilities, seasonal changes) and it is not possible to attribute changes observed in this pilot study (in either direction) to the leash walking training. Using a randomized, controlled design in future studies will help ensure observed changes can be attributed to the intervention. A sufficiently large sample will also allow for statistical control of some confounding variables.

The study sample was homogenous (mostly White and female), young (mostly in 60s), and highly active (average steps per day at baseline was nearly 7800). The University where this study was conducted is located in a county in Western Massachusetts where a large majority of residents identify exclusively as White (88% according to the U.S. Census Bureau [42]). Although recruitment efforts reached residents in neighboring counties with greater racial diversity, the travel requirements of the study may have prohibited participation from people who lived farther from the University. Future studies could consider offering the intervention classes at off-campus sites in towns or cities with more racially and ethnically diverse residents. Majority female samples are a common issue in human–animal bond research studies [43]. If this intervention appeals more to females, future studies could focus on females-only, especially since females in the U.S. are less active than males and therefore in greater need of PA intervention [44]. Our strict study exclusion criteria regarding fall risk may have led older and less active older adults to be deemed ineligible. Future studies testing this intervention approach should consider allowing higher risk individuals to participate if cleared by their physician, as these participants have the most to gain from becoming more physically active. Additional measures could be taken to further reduce fall risk, such as including a strength and balance training component and tips for preventing dog walking-specific falls (e.g., avoid non-retractable leashes). 

The primary strength of this study is the novel intervention approach tested. We focused on promoting dog walking, a purposeful form of PA that lends itself well to habit formation and can increase social interaction and time outdoors. Our approach targeted established correlates of dog walking (strength of the dog-owner bond) and PA more broadly (enjoyment) and investigated an intervention (leash walking skills training) that already exists in many communities but could be better promoted and made more accessible to older adults. Notably, the dog trainer who led the leash walking skills training course in this study did not modify her course in any way to meet study goals. Participants received the same leash walking skills training they would have received if they signed up for the course in the community rather than through our research study. The only additional content they received was information on PA benefits and tips for establishing a dog walking routine. We purposefully kept this content simple and brief to ensure it could easily be manualized and delivered by a non- PA expert in future studies. Future studies could also consider testing loose leash walking training with and without the additional PA behavior content to see if the additional content is needed. Long-term, if this approach is deemed effective at increasing PA and improving other health outcomes among older adults, our vision is that aging councils, senior centers, or other community organizations who support healthy aging could encourage older adult dog owners to enroll in loose leash walking training, and work with local dog trainers to offer classes just for seniors and/or offer seniors discounted rates. The evidence-based PA content (if determined to add value) could be made available in a printed handout or YouTube link for dog trainers to provide to older adults who enroll in their leash walking training courses. Community organizations could even facilitate partnerships between dog trainers and fitness professionals who could offer complementary programming to reduce dog walking fall risk (e.g., dog trainers could provide information on local strength and balance training classes).

We hope this feasibility study inspires others to build off our work, or to develop and test other interventions that leverage the human–animal bond to support physical activity in older adulthood. We feel behavioral dog walking interventions align well with newer PA theoretical models and frameworks, such as the Multi-Process Action Control (M-PAC) Framework [22,23,24,25], which include traditional social cognitive constructs but also emphasize enjoyment and the role of non-conscious regulatory processes (e.g., habit, identity, automatic associations with PA) in PA. Dog walking (like all forms of PA) has multiple levels of influence, and environmental and policy interventions guided by the social ecological approach are also needed to support and encourage dog walking among older adults [45]. Creating supportive physical environments for dog walking would benefit dog walkers of all ages. Policy-level interventions could also move upstream and address barriers to dog ownership for older adults (e.g., pet-restrictions in senior housing facilities).

Given high rates of dog ownership among older Americans [46], behavioral health interventions that leverage the dog-owner bond have potential for high-level reach in this population. This study demonstrated that leash manners training is a feasible and acceptable PA intervention approach with community-dwelling older adults. A controlled study with a more diverse, less physically active sample is warranted. 

## Figures and Tables

**Figure 1 geriatrics-07-00120-f001:**
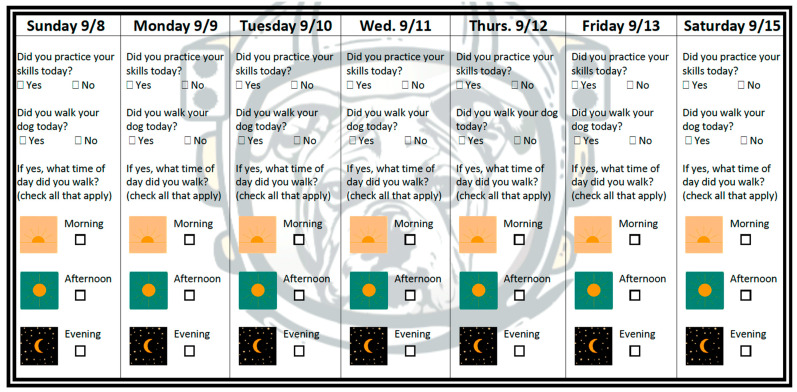
Study tracking sheet.

**Figure 2 geriatrics-07-00120-f002:**
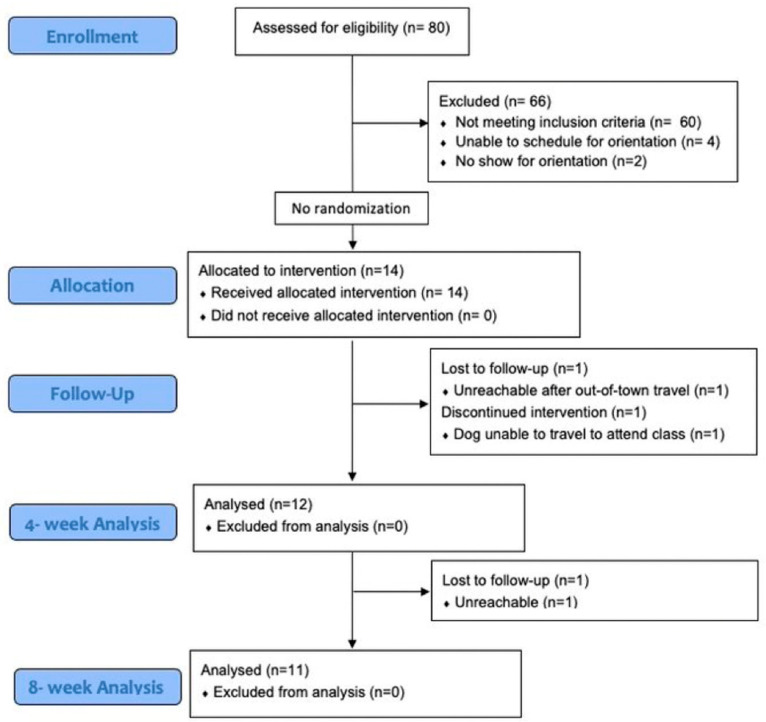
Consort flow diagram.

**Figure 3 geriatrics-07-00120-f003:**
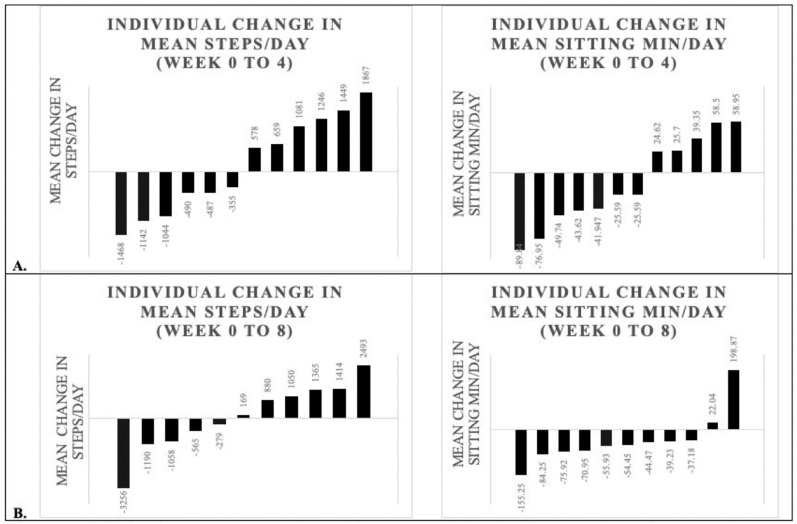
**Panel A**: Individual participant changes in steps/day and sitting minutes/day from baseline to 4 weeks (n = 12). **Panel B**: Individual participant changes in steps/day and sitting minutes/day from baseline to 8 weeks (n = 11). Data collected using activPAL activity monitors. Note: Baseline data excludes two participants who did not complete the program. Dog walking enjoyment is missing one additional participant at each time point.

**Table 1 geriatrics-07-00120-t001:** Participant Characteristics (*n*=14).

Age (*M; SD*)	64.9 (4.3)
60–64 *n* (%) 65–69 *n* (%) 70–74 *n* (%) 75–80 *n* (%)	7 (50.0%) 5 (35.7%) 1 (7.1%) 1 (7.1%)
Female *n* (%)	10 (71.4%)
BMI (M; SD)	31.7 (7.6)
Normal Overweight Obese	2 (14.3%) 2 (14.3%) 10 (71.4%)
Education *n* (%)	
High school 2–4-year degree Graduate/Prof. Degree Other	1 (7.1%) 7 (50.0%) 4 (28.6%) 2 (14.3%)
Race / Ethnicity *n* (%)	
White/European	13 (92.9%)
Employment *n* (%)	
Retired / Unemployed Part-time Full-time	8 (57.1%) 3 (21.4%) 3 (21.4%)
Income	
<$40,000 $40,000–100,000 >$100,000 Preferred not to answer	4 (28.6%) 3 (21.4%) 3 (21.4%) 4 (28.6%)
Marital Status *n* (%)	
Married Unmarried/Divorced/Widowed	7 (50.0%) 7 (50.0%)
Living Situation	
Not alone Alone	11 (78.6%) 3 (21.4%)
Community *n* (%)	
Suburban Rural Urban	6 (42.9%) 6 (42.9%) 2 (14.3%)
SF-36	
General Health *n* (%)	
Very Good Good Fair	7 (50.0%) 6 (42.9%) 1 (7.1%)

**Table 2 geriatrics-07-00120-t002:** Physical activity, dog walking, and psychosocial outcomes.

	Baseline Mean (SD)	4 wk Mean (SD)	8 wk Mean (SD)	Baseline to 4 wk Mean (SD) Change	Baseline to 8 wk Mean (SD) Change
	n = 12	n = 12	n = 11	n = 12	n = 11
Steps/day—activPAL	7815.8 (2024.6)	7973.6 (2415.6)	8147.7 (2865.7)	157.8 (1125.3)	93.0 (1594.4)
Sitting minutes/day- activPAL	1114.8 (88.9)	1107.8 (99.2)	1065.6 (107.6)	−7.0 (53.3)	−36.1 (88.7)
Dog walk days/week	3.3 (2.5)	4.1 (1.9)	2.9 (2.2)	0.8 (1.2)	−0.3 (1.2)
Total # dog walks/week	4.7 (4.8)	5.5 (4.3)	3.7 (3.6)	0.8 (1.7)	−0.9 (2.4)
	n = 12	n = 11	n = 11	n = 11	n = 11
Dog-owner relationship ^a^	38.5 (0.6)	38.0 (5.7)	37.9 (5.8)	0.0 (2.6)	−0.1 (2.0)
Dog walking enjoyment ^b^	96.4 (20.4)	103.0 (12.3)	101.8 (14.6)	4.2 (9.1)	3.8 (11.8)
Dog walking automaticity ^c^	_	11.6 (3.6)	11.6 (3.7)	_	_

^a^ Dog-owner relationship measured with Comfort from Companion Animals Scale; scores range from 0–44 with higher scores indicating greater attachment. ^b^ Dog walking enjoyment measured with modified Physical Activity Enjoyment Scale; scores range from 18–126 with higher scores indicating greater enjoyment. ^c^ Dog walking automaticity measured with four questions from the Self-Reported Habit Index; scores range from 4–20 with higher scores indicating stronger automaticity. SD = standard deviation.

## Data Availability

The data presented in this study are available on request from the corresponding author.

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
