# Peer review of "Testing Leash Walking Training as a Physical Activity Intervention for Older Adult Dog Owners: A Feasibility Study"

_geriatrics, 2022, doi:10.3390/geriatrics7060120_

Round 1
Reviewer 1 Report
Review of Potter, Rajala, Chase, and LeBlanc's Can Leash Walking Training Increase Physical Activity Among Older Dog Owners?, manuscript ID geriatrics-1956566
This manuscript addresses a potential way of increasing the physical activity level of older people who own dogs. Increasing physical activity is an important issue in this population and can lead to substantial improvements in health and well-being. Thus, the manuscript appears to be appropriate for this journal and has the potential to make a meaningful contribution to the literature. The manuscript is clear and concisely written and is free of typographical and grammatical mistakes.
The basic research design is a pre-test, post-test design with a follow-up four weeks later. As the authors note, such designs are inherently difficult to interpret because of the possibility of confounds when there is no equivalent control group. While taking physical activity measures for seven days at each measurement period helps to reduce some potential confounds, the confounds may still be present in such a design. The authors do not adequately acknowledge the factors other than the treatment that might influence how active the participants were on each of these days – weather (was it rainy and cold at baseline but sunny and warm at follow-up), number and type of appointments (did they have to wait a long time at a doctors appointments at baseline but not at follow-up), activities (did they attend church services involving a couple hours of sitting during baseline but not at follow-up), health (did they feel poorly at baseline but not at follow-up), etc. Because of these and other such factors, it is difficult, at best, to make meaningful comparisons between baseline and post-treatment and follow-up.
While the authors acknowledge that the study is a preliminary investigation and therefore has a very small sample, it is far from clear that the sample represents the population. The sample seems biased to obese, white, well-educated females. This is acknowledged in the discussion, but such a sample limits the authors' ability to generalize from the sample to the population and thus limits the manuscript's contribution to the literature.
Figure 3 shows that there is a lot of variability in the mean change in steps per day and sitting minutes per day from weeks 0 to 4 and, to a lesser extent, from weeks 0 to 8. Approximately half of the participants had a decrease in the number of steps per day and an increase in the sitting minutes per day from week 0 to 4 – the opposite of what was predicted. That is, if the authors claim that the treatment helps increase the physical activity level of some participants, then the treatment seems to decrease physical activity in others. It appears that the program helps some people but hurts some people to an almost equal extent (steps per day) or to a lesser extent (sitting minutes per day). This is not explicitly mentioned in the discussion and is a major limit on the manuscript's contribution to the literature. It would be helpful if the authors used data or discussed hypotheses about why the program increases physical activity in some people but decreases it to nearly the same extent in others.
This decrease in physical activity in some participants is as important as the increase in physical activity in other participants. This needs to be made more explicit and clearer in the title, abstract, and discussion. The title tends to hide the decrease in physical activity. Rather than wording the title as a question, it would be more appropriate and informative to word it as a statement – Leash Walking Training Increases Physical Activity Among Some Older Adult Dog Owners but Decreases It Among Others.
Given the preliminary investigational nature of the study, there are no inferential statistics. Given the sample size and variability in the data, inferential statistics probably would not be statistically significant. Thus, one cannot really tell whether the differences are likely due to the treatment or are what one would expect by chance. This further reduces the contribution the manuscript can make to the literature. I encourage the authors to consider reporting effect sizes for their findings. Effect sizes do not depend on sample size but would indicate whether any observed effect is sufficiently large to warrant further consideration in future research. This might or might not, given the size of the effects, increase the contribution the manuscript can make to the literature.
Finally, because the treatment included a presentation on the health benefits of physical activity, guidelines on physical activity, the goals of the study, strategies for making dog walking a habit, it is difficult to claim that it is the leash walking training and not these other topics that lead to any change in the physical activity levels of the participants. While I agree that the leash walking training likely played some, perhaps a large, role in the observed effects, the authors are not entitled to claim, that it is the leash walking training per se that led to the effects.

Author Response
This manuscript addresses a potential way of increasing the physical activity level of older people who own dogs. Increasing physical activity is an important issue in this population and can lead to substantial improvements in health and well-being. Thus, the manuscript appears to be appropriate for this journal and has the potential to make a meaningful contribution to the literature. The manuscript is clear and concisely written and is free of typographical and grammatical mistakes.
The basic research design is a pre-test, post-test design with a follow-up four weeks later. As the authors note, such designs are inherently difficult to interpret because of the possibility of confounds when there is no equivalent control group. While taking physical activity measures for seven days at each measurement period helps to reduce some potential confounds, the confounds may still be present in such a design. The authors do not adequately acknowledge the factors other than the treatment that might influence how active the participants were on each of these days – weather (was it rainy and cold at baseline but sunny and warm at follow-up), number and type of appointments (did they have to wait a long time at a doctors appointments at baseline but not at follow-up), activities (did they attend church services involving a couple hours of sitting during baseline but not at follow-up), health (did they feel poorly at baseline but not at follow-up), etc. Because of these and other such factors, it is difficult, at best, to make meaningful comparisons between baseline and post-treatment and follow-up.
Thank you for this constructive feedback. This is in line with the feedback from the academic editor, which asked that we revise the manuscript to focus fully on feasibility aspects of the study and refrain from making any conclusions about the effects of the intervention. We have removed content from the abstract and manuscript body that implied the intervention led to improvements in some participants. We now stress in multiple locations in the Discussion section that the possibility of uncontrolled confounding prohibits our ability to draw conclusions about even preliminary intervention effectiveness.
While the authors acknowledge that the study is a preliminary investigation and therefore has a very small sample, it is far from clear that the sample represents the population. The sample seems biased to obese, white, well-educated females. This is acknowledged in the discussion, but such a sample limits the authors' ability to generalize from the sample to the population and thus limits the manuscript's contribution to the literature.
After removing content around preliminary intervention effectiveness, we now spend more time in the Discussion talking about how future studies may be able to enroll more diverse study participants.
Figure 3 shows that there is a lot of variability in the mean change in steps per day and sitting minutes per day from weeks 0 to 4 and, to a lesser extent, from weeks 0 to 8. Approximately half of the participants had a decrease in the number of steps per day and an increase in the sitting minutes per day from week 0 to 4 – the opposite of what was predicted. That is, if the authors claim that the treatment helps increase the physical activity level of some participants, then the treatment seems to decrease physical activity in others. It appears that the program helps some people but hurts some people to an almost equal extent (steps per day) or to a lesser extent (sitting minutes per day). This is not explicitly mentioned in the discussion and is a major limit on the manuscript's contribution to the literature. It would be helpful if the authors used data or discussed hypotheses about why the program increases physical activity in some people but decreases it to nearly the same extent in others.
We appreciate this feedback. We have added content in the Discussion section explicitly stating that some participants improved their physical activity and sedentary behavior, while others took fewer steps and sat more minutes per day after the intervention. We stress that changes in either direction cannot be attributed to the intervention given the limits of the study design, and point to a number of potential confounding factors. We also note that future studies should use larger samples and randomized designs to limit the risk of confounding.
This decrease in physical activity in some participants is as important as the increase in physical activity in other participants. This needs to be made more explicit and clearer in the title, abstract, and discussion. The title tends to hide the decrease in physical activity. Rather than wording the title as a question, it would be more appropriate and informative to word it as a statement – Leash Walking Training Increases Physical Activity Among Some Older Adult Dog Owners but Decreases It Among Others.
As the academic editor asked us to focus the revised manuscript fully on feasibility aspects of the study, we chose not to highlight that physical activity increased in some participants and decreased in others in the title and abstract. As noted in a previous response, we now explicitly state in the Discussion that some participants improved their physical activity and sedentary behavior, while others took fewer steps and sat more minutes per day after the intervention. We also clearly state that these changes (positive or negative) cannot be attributed to the intervention due to limitations of the study design. We agree that the title should be modified to be more transparent, and it now reads: Testing Leash Walking Training as a Physical Activity Intervention for Older Adult Dog Owners: A Feasibility Study.
Given the preliminary investigational nature of the study, there are no inferential statistics. Given the sample size and variability in the data, inferential statistics probably would not be statistically significant. Thus, one cannot really tell whether the differences are likely due to the treatment or are what one would expect by chance. This further reduces the contribution the manuscript can make to the literature. I encourage the authors to consider reporting effect sizes for their findings. Effect sizes do not depend on sample size but would indicate whether any observed effect is sufficiently large to warrant further consideration in future research. This might or might not, given the size of the effects, increase the contribution the manuscript can make to the literature.
We carefully considered the suggestion to report effect sizes in this manuscript. Ultimately, we decided not to include them, as we felt their inclusion would conflict with the manuscript’s full focus on feasibility and consistent message that attempting to draw any conclusions about program effectiveness is inappropriate due to study design limitations.
Finally, because the treatment included a presentation on the health benefits of physical activity, guidelines on physical activity, the goals of the study, strategies for making dog walking a habit, it is difficult to claim that it is the leash walking training and not these other topics that lead to any change in the physical activity levels of the participants. While I agree that the leash walking training likely played some, perhaps a large, role in the observed effects, the authors are not entitled to claim, that it is the leash walking training per se that led to the effects.
Thank you for this feedback. We feel that the revised manuscript sends a clear, consistent message that we are refraining from drawing any conclusions about intervention effectiveness due to the small sample size and single-arm design. We agree that future studies testing this program should consider whether any effects are driven by the leash walking training or the behavior change content, and perhaps even compare programs with and without the behavioral content to parse this out. We added a sentence in the Discussion. (lines 392-393) that includes this suggestion for future research.
Reviewer 2 Report
Dear authors,
Congratulations on the completion of this research work, for your time and dedication.
My comments are very positive about your research.
I recommend some suggestions, which will improve your manuscript, downloads, recent citations and increase internationalisation:
Conceptualise the context of the benefits of physical activity practice better with current citations to the readers, as the introduction has fallen short and you need to contextualise your study:
-In the first paragraph of your introduction, in line 32, when you mention that the practice of physical activity helps in physical and cognitive function, you need to elaborate a bit more. I suggest that you add "A recent study in older people indicates that physical activity helps older adults to maintain their functional abilities and leads to improved life satisfaction" https://doi.org/10.3390/socsci11060265
-In the first paragraph of your introduction, when you talk about the practice of physical activity, I advise you to mention variables that are beyond our reach and that can influence the practice of activity. I point you to a study to incorporate it and better contextualise the situation, in line 36 of your document "A recent study indicates that regular physical activity practice in people over 60 is associated with some socio-environmental and contextual levels, such as higher educational level and income, so attention should be paid to variables that are beyond our reach and that can influence to better understand the motives and variables that influence the practice of physical activity in older people" DOI: 10.3390/ijerph182010815
-There is a special article, which does not mention the use of dogs, but in the introductory part, at the end of the first paragraph, it can help you to contextualise the introduction much better, since it says "In a study that has investigated predictors of the level of physical activity in older people, it indicates that gender, education, activity and leisure and health are the most important predictors of physical activity in older people, activity and leisure and health are predictors of the practice of physical activity, being necessary the use of leisure activities such as playing, visiting friends an indirect way to increase physical activity, being necessary to know the characteristics of the environment that indicate a higher level of physical activity in older people" https: //www. mdpi.com/2076-328X/12/9/331/htm
-On the methods and procedure, you have described them very well and give the possibility to replicate the study.
-Towards the end of your discussion I suggest you incorporate a few paragraphs in response:
a) what theoretical implications does this work have for the scientists who read this work, for theorists in the field or colleagues?
b) strengths of your work with respect to other studies.
c) what practical implications does this work have for older people?
d) future line of research derived from this work and that needs to be covered.
This is a new topic in the field of the elderly, so there are few studies to support the research.
I hope it will get better visibility this way! Congratulations, I loved your work!
My sincere congratulations for the work.
Author Response
Dear authors,
Congratulations on the completion of this research work, for your time and dedication.
My comments are very positive about your research.
We want to thank the reviewer for their kind and positive feedback on this feasibility work.
I recommend some suggestions, which will improve your manuscript, downloads, recent citations and increase internationalisation:
Conceptualise the context of the benefits of physical activity practice better with current citations to the readers, as the introduction has fallen short and you need to contextualise your study:
-In the first paragraph of your introduction, in line 32, when you mention that the practice of physical activity helps in physical and cognitive function, you need to elaborate a bit more. I suggest that you add "A recent study in older people indicates that physical activity helps older adults to maintain their functional abilities and leads to improved life satisfaction" https://doi.org/10.3390/socsci11060265
Thank you for this suggestion. We added a sentence beginning on line 33 to add that maintaining health and functional ability are key predictors of life satisfaction in older adults. We cite the suggested paper.
-In the first paragraph of your introduction, when you talk about the practice of physical activity, I advise you to mention variables that are beyond our reach and that can influence the practice of activity. I point you to a study to incorporate it and better contextualise the situation, in line 36 of your document "A recent study indicates that regular physical activity practice in people over 60 is associated with some socio-environmental and contextual levels, such as higher educational level and income, so attention should be paid to variables that are beyond our reach and that can influence to better understand the motives and variables that influence the practice of physical activity in older people" DOI: 10.3390/ijerph182010815
Thank you for this helpful suggestion. We have added a sentence beginning on line 37 that recognizes that certain factors beyond the reach of health behavior interventionists affect physical activity levels in older adults (e.g., income, education level) and included the suggested citation.
-There is a special article, which does not mention the use of dogs, but in the introductory part, at the end of the first paragraph, it can help you to contextualise the introduction much better, since it says "In a study that has investigated predictors of the level of physical activity in older people, it indicates that gender, education, activity and leisure and health are the most important predictors of physical activity in older people, activity and leisure and health are predictors of the practice of physical activity, being necessary the use of leisure activities such as playing, visiting friends an indirect way to increase physical activity, being necessary to know the characteristics of the environment that indicate a higher level of physical activity in older people" https: //www. mdpi.com/2076-328X/12/9/331/htm
We agree with the reviewer (and the authors of the suggested paper) that encouraging leisure activities such as visiting friends is a great ‘indirect’ way to increase physical activity in older people, and that encouraging dog walking might be another way to increase physical activity indirectly in this population. At the same time, the intervention tested in this study was very direct in its efforts to increase physical activity. It was advertised as a physical activity intervention, and it included content on the U.S. physical activity guidelines and health benefits of physical activity. Therefore, we respectfully decline the suggestion to include this content and citation in our Introduction.
-On the methods and procedure, you have described them very well and give the possibility to replicate the study.
Thank you!
-Towards the end of your discussion I suggest you incorporate a few paragraphs in response:
- a) what theoretical implications does this work have for the scientists who read this work, for theorists in the field or colleagues?
- b) strengths of your work with respect to other studies.
- c) what practical implications does this work have for older people?
- d) future line of research derived from this work and that needs to be covered.
Thank you for the helpful suggestions for how to strengthen our Discussion section. We have added two in-depth paragraphs at the end of the Discussion section. The first paragraph highlights the primary strength of our study (the novel intervention approach) and the practical implications of this research for older adults and communities if future studies find the approach to be effective at increasing physical activity. The second paragraph discusses this line of research in the context of contemporary physical activity theories and models, and suggests that future research test interventions that support dog walking at multiple levels (behavioral, environmental, policy).
This is a new topic in the field of the elderly, so there are few studies to support the research.
I hope it will get better visibility this way! Congratulations, I loved your work!
My sincere congratulations for the work.
Thank you again for your kind and supportive feedback on this work!
Round 2
Reviewer 1 Report
The authors have adequately addressed most of my concerns. However, most of the means are small relative to the variability in the data and often 0 is included in the range. This implies that the effects in the population may be very small if they exist at all. While the treatment may be well tolerated by their non-representative sample, there is no clear evidence that the treatment is effective. If there is no clear evidence that the treatment is effective, why would one use the treatment in future research?